# *S. aureus* alpha-toxin monomer binding and heptamer formation in host cell membranes – Do they determine sensitivity of airway epithelial cells toward the toxin?

**Nils Möller**[1], **Sabine Ziesemer**[1], **Petra Hildebrandt**[2], **Nadine Assenheimer**[1], **Uwe Völker**[2], **Jan-Peter Hildebrandt**[1] *

**1** Animal Physiology and Biochemistry, University of Greifswald, Greifswald, Germany, **2** Interfaculty Institute for Genetics and Functional Genomics, University Medicine Greifswald, Greifswald, Germany

* jph@uni-greifswald.de

**Data Availability Statement:** Original data are deposited at BioStudies (EMBL-EBI) (https://www.

## Abstract

Alpha-toxin (Hla) is a major virulence factor of *Staphylococcus aureus* (*S. aureus*) and plays an important role in *S. aureus*-induced pneumonia. It binds as a monomer to the cell surface of eukaryotic host cells and forms heptameric transmembrane pores. Sensitivities toward the toxin of various types of potential host cells have been shown to vary substantially, and the reasons for these differences are unclear. We used three human model airway epithelial cell lines (16HBE14o-, S9, A549) to correlate cell sensitivity (measured as rate of paracellular gap formation in the cell layers) with Hla monomer binding, presence of the potential Hla receptors ADAM10 or α5β1 integrin, presence of the toxin-stabilizing factor caveolin-1 as well as plasma membrane lipid composition (phosphatidylserine/choline, sphingomyelin). The abundance of ADAM10 correlated best with gap formation or cell sensitivities, respectively, when the three cell types were compared. Caveolin-1 or α5β1 integrin did not correlate with toxin sensitivity. The relative abundance of sphingomyelin in plasma membranes may also be used as a proxi for cellular sensitivity against alpha-toxin as sphingomyelin abundances correlated well with the intensities of alpha-toxin mediated gap formation in the cell layers.

## Introduction

The human respiratory epithelium and its luminal mucus layer constitute the barrier between the airspace and the interior of the body [1]. Secretory cells export mucins, salt and water which results in formation of the airway surface liquid (ASL). It consists of two layers, a thin (approx. 8 μm) periciliary liquid layer (PCL) of low viscosity in which the cilia of the ciliated cells of the airway epithelium beat, and a thicker mucus layer (up to 50 μm) above the PCL that is highly viscous and serves to trap inhaled dust particles and pathogens [2]. By the beating of the cilia the mucus layer including the adherent particles is transported towards the pharynx

ebi.ac.uk/biostudies/) under the accession number S-BSST397.

**Funding:** We acknowledge support for the Article Processing Charge from the DFG (German Research Foundation, 393148499) and the Open Access Publication Fund of the University of Greifswald. Nils Möller is the recipient of a graduate student stipend of the State of Mecklenburg-Vorpommern, Germany. The funders had no role in study design, data collection and analysis, decision to publish, or preparation of the manuscript.

**Competing interests:** The authors have declared that no competing interests exist.

at a speed of approximately 60 μm/s where the material is usually swallowed. This process is termed mucociliary clearance and keeps our airways free of potentially harmful agents [2].

In healthy people, an inhaled particle is usually disposed off within 2 h so that pathogens are unable to form colonies or biofilms at the surface of the mucus layer or to get in touch with the apical cell surfaces of the epithelial cells [3]. However, if the mucociliary clearance is disturbed as in patients with cystic fibrosis or bedridden people, pathogens such as the commensal and opportunistic bacterium *S. aureus* and others remain much longer in the airways, may become permanent residents and reach higher densities [4, 5]. This condition may trigger expression of virulence-associated factors [6, 7]. These molecules are supposed to be much more mobile in the mucus layer than the bacteria and may readily reach the apical surface of the epithelial cell layer [3].

The major virulence-factor involved in the lung pathogenicity of *S. aureus* is alpha-toxin (haemolysin A, Hla), a pore forming toxin that is secreted by the bacteria as a 33 kDa monomer [8]. At low concentrations the monomers bind to specific plasma membrane receptors of eukaryotic host cells [9], whereas at high concentrations (> 1 μmol/l) they can also bind non-specifically to phosphocholine headgroups of phospholipids like sphingomyelin or phosphatidylcholine of the plasma membrane [9–11].

Several types of plasma membrane proteins on the surface of different types of potential host cells are supposed to serve as toxin receptors, including metalloproteinase ADAM10 (a disintegrin and metalloproteinase 10) [12, 13], α5β1 integrin [14, 15] or anion exchanger 1 (AE1 or band 3 protein) [16]. The band 3 protein may be a toxin receptor in erythrocytes [16]. Another plasma membrane protein with binding ability for alpha-toxin is caveolin-1 although it does not have a domain that protrudes into the extracellular space [17]. This protein may play a role in the stabilization of the toxin in the plasma membrane upon binding [18].

Plasma membrane bound alpha-toxin monomers form a heptameric pre-pore that is firmly attached to the cell surface but is still non-lytic [19]. Especially in the presence of phospholipids containing choline headgroups (phosphatidylcholine and sphingomyelin) that, together with cholesterol, form chemically unique domains (lipid rafts) in the plasma membranes of eukaryotic cells [20], the pre-pores may quickly form functional transmembrane pores. During that process each of the heptamers rolls out a domain comprised of two beta-sheets that penetrates the plasma membrane. Together, these domains form an aqueous transmembrane channel (beta-barrel) [19, 21]. Areas in the plasma membrane of cells that have a high content of sphingolipids and cholesterol are important sites for signal transduction and endocytosis [22, 23]. They may also have an important function in mediating pore-formation of bacterial toxins like staphylococcal alpha-toxin [24]. This has been confirmed by a recent study showing that pore-formation is completely suppressed in cell membranes that had been depleted of sphingomyelin [25].

The alpha-toxin transmembrane pore is permeable for different cations like $Na^+$, $K^+$ or $Ca^{2+}$ [8, 26–28] and even for small organic molecules like ATP [29]. In airway epithelial cells, this results in changes in membrane potential, cytosolic ion concentrations, cell signaling, actin cytoskeleton architecture and ultimately in the loss of cell-cell and cell-matrix contacts which results in the formation of paracellular gaps in the epithelial cell layer [30–32]. *In vivo*, such effects of pore-formation may disrupt the barrier function of the respiratory epithelium [33].

Various epithelial cell types exposed to alpha-toxin have been observed to show largely different sensitivities. The immortalized human airway epithelial cell line S9 can cope with high concentrations of alpha-toxin (2,000 ng/ml) while other immortalized cells (16HBE14o-) or lung cancer cells (A549) are massively damaged at the same concentration [31, 34]. Aim of this study was to elucidate whether the differences in toxin sensitivities in these three cell types are due to different rates of toxin binding to plasma membrane receptors and/or to differences in

pore-formation and pore abundance due to different interactions of the pore complex with plasma membrane proteins or lipids.

## Materials and methods

### Chemicals and reagents

Trypsin (including ethylene diamine tetraacetic acid, EDTA) (L2143) was purchased from Biochrom (Berlin, Germany). Accutase (including EDTA) (P10-21500) was obtained from PAN-Biotech (Aidenbach, Germany). Sphingomyelin (85615), phosphatidylcholine (P3556), phosphatidylserine (P0474), phosphatidylethanolamine (P7693) and sphingomyelinase from *Bacillus cereus* (S9396) were ordered from Sigma (Steinheim, Germany). WesternBright chemiluminescence substrate from Advansta (K-12045-D50) was purchased from Biozym (Oldendorf, Germany). Trypsin inhibitor from soybeans (A1828,0005) was obtained from Applichem (Darmstadt, Germany).

Antibodies (Ab) were obtained from these sources: Hla-Ab (S7531) from Sigma (Steinheim, Germany); ADAM10 Ecto (MAB1427-100) from R&D Systems and purchased through anti-koerper-online.de (Aachen, Germany); Caveolin-1 (7C8) (sc-53564), normal mouse IgG$_{2b}$ (sc-3879) from Santa Cruz Biotechnology (Heidelberg, Germany); Integrin α5β1 (M200) (NBP2-52680) from Novus Biologicals and purchased through Bio-Techne (Wiesbaden, Germany); Alexa Fluor$^{®}$ 594 AffiniPure goat anti-mouse IgG (H+L) (115-585-003) from Jackson ImmunoResearch and purchased through Dianova (Hamburg, Germany); goat anti-rabbit IgG-HRP (7074s) and anti-rabbit IgG (H+L) F(ab')$_2$ Fragment Alexa Fluor$^{®}$ 594 Conjugate (8889S) from Cell Signaling (Frankfurt am Main, Germany). All other chemicals were reagent grade and obtained from Roth (Karlsruhe, Germany).

### Expression and purification of recombinant *S. aureus* alpha-toxin (rHla) and enhanced green fluorescent protein coupled rHla (rHla-eGFP)

Recombinant alpha-toxin (rHla) was prepared and purified as described previously [35]. The plasmid for the preparation of rHla-eGFP was designed by Dr. Christian Müller (University of Greifswald, Germany) and this fusion protein was produced exactly like Hla. Purity of the toxins was verified by SDS-PAGE and Coomassie brilliant blue staining. The protein concentration of the toxins was determined using the Bradford assay [36]. Biological activities of rHla as well as rHla-eGFP were tested in a haemolysis assay in sheep blood agar. Aliquots of rHla and rHla-eGFP were stored at −80˚C or in the vapor phase of liquid nitrogen. An rHla concentration of 1,000 ng/ml (30 nmol/l) was routinely used to avoid any non-specific binding of the toxin to the lipid environment of the host cell membranes [9].

### Human airway model epithelial cell cultures and culture conditions

Two immortalized human airway epithelial cell lines (16HBE14o-, S9) and one alveolar cancer cell line (A549) were used for the experiments. With permission of D.C. Gruenert 16HBE14o-cells were obtained from K. Kunzelmann (University of Regensburg, Germany). S9 cells were purchased from ATCC-LGC Standards (Wesel, Germany, S9). A549 cells were obtained from the cell collection of the Friedrich Loeffler-Institute (Riems, Germany). References describing generation and characteristics of these cell lines are listed in Supplement 1.

Cells were cultured on 10 cm Cell+ dishes (3902300, Sarstedt, Numbrecht, Germany) in Eagle's MEM (PAN-Biotech, Aidenbach, Germany) containing 10% FBS superior (S 0615, Biochrom, Berlin, Germany), 29,8 mmol/l NaHCO$_3$ and 1% penicillin/streptomycin solution (A 2213, Biochrom, Berlin, Germany) (final concentration: 100 μg/l) at 37˚C and gassing with 5%

$CO_2$. Cell culture medium of A549 cells additionally contained 1% L-glutamine solution (K 0283, Biochrom, Berlin, Germany) (final concentration: 2 mmol/l). Medium was changed every 3–4 days. Shortly before cells formed confluent monolayers, they were passaged on new cell culture dishes or directly used in the experiments. To passage the cells, the cell culture medium was removed and the cells were washed with 10 ml PBS. Subsequently, 1 ml trypsin + EDTA were applied to the cell layer and the dish was incubated at 37˚C until the cells detached from the cell culture dish. Cells were resuspended several times to separate them from each other and then transferred to a new 10 cm cell culture dish at a diluton of 1: 5. All cell cultures were checked for *Mycoplasma* contaminations on a regular basis.

## Time lapse microscopy

All three airway model epithelial cell lines (16HBE14o-, S9, A549) were cultured in 35 mm μ-cell culture plates (Ibidi, Planegg, Germany) in cell culture medium and under the conditions described above until they reached 90 – 100% confluency. One day before cells were treated with 1,000 ng/ml rHla or PBS (control) and placed under the time lapse-microscope (Biosta-tion II, Nikon Instruments, Düsseldorf, Germany) the cell culture medium was renewed. During the experiments the microscope chamber was kept at 37˚C and under a $CO_2$ (in air) concentration of 5%. Images of cells were taken every 3 min over 24 h and combined to time lapse-movies of 60 s duration. To check the sensitivity of the cells to the toxin at a concentration of 1,000 ng/ml, the sum of the areas of gaps in rHla treated cells were measured using Fiji (Fiji Is Just ImageJ) [37].

## Sample preparation for Western blotting

As optimized in preliminary experiments, cells were treated with 1,000 ng/ml of recombinant Hla or with phosphate buffered saline (PBS) (negative control) for 1 h.

After incubation with rHla or PBS (negative control), the culture medium was carefully aspirated and the cells were washed using $2 \times 5$ ml PBS. Then 400 μl lysis buffer (100 mmol/l KCl, 20 mmol/l NaCl, 2 mmol/l $MgCl_2$, 0.96 mmol/l $NaH_2PO_4$, 0.84 mmol/l $CaCl_2$, 1 mmol/l EGTA, 0.5% (v/v) Tween 20, 25 mmol/l HEPES (free acid), pH 7.2 containing aprotinin (0.31 mmol/l), leupeptin (4.21 μmol/l) and pepstatin (2.92 μmol/l), as well as 1 mmol/l PMSF and 0.33 mmol/l ortho-vanadate) was added to each of the 10 cm dishes of cultured cells. The cells were subsequently scraped off the cell culture dish and the suspensions were transferred to 2 ml reaction tubes and homogenized for 30 s on ice using a T8-Ultraturrax (IKA Labortechnik, Staufen, Germany). A small portion of homogenate was used for determination of protein concentration [36]. The remaining volume of the homogenate was mixed with an equal volume of SDS sample-buffer, aliquoted and frozen at -80˚C until use.

## Semi-quantitative Western blotting

Samples (15 μg total protein) were separated by SDS/PAGE (10% gel) in a minigel apparatus (BioRad, Munich, Germany) and transferred onto nitrocellulose membrane (HP40, Roth, Karls-ruhe, Germany) by wet blotting. Nitrocellulose membranes were wetted in a buffer containing SDS (48 mmol/l Tris (free base), 39 mmol/l glycine, 20% methanol [v/v], 1.3 mmol/l SDS, pH 9.2). Gels with separated proteins were soaked in transfer buffer with 10% methanol (v/v) and without SDS for approx. 20 min. Blotting was performed in a wet blotting system (BioRad, Munich, Germany) at 100 V for 1 h and at 4˚C in transfer buffer containing 10% methanol.

Quantification of rHla mono- and heptamers was performed by Western blotting using a Hla primary antibody (1:3333), a goat anti-rabbit IgG-HRP secondary antibody (1:6,000) and a chemiluminescence substrate (Advansta WesternBright, Biozym, Hess. Oldendorf,

Germany). Chemiluminescent signals were detected using an Intas Chemostar ECL imager (Intas, Göttingen, Germany). Phoretix 1 D was used for the densitometric measurement of the protein bands (Nonlinear Dynamics, Newcastle upon Tyne, UK). To correct for potential minor differences in exposure time, the mean density of all bands on each membrane image was used to normalize the densities of individual bands of the same membrane. In case of comparisons of different cell lines, signal intensities of rHla bands were not normalized to band intensities of housekeeping proteins such as β-actin because of potential differences in the content of such proteins. These results were instead expressed as signal intensities of Hla bands in relation to the total protein content of the samples.

### rHla-eGFP binding assays

In preliminary experiments, we verified that the fusion protein of rHla with eGFP was able to bind to cell surfaces of airway epithelial cells and form functional transmembrane pores. These findings encouraged us to perform binding assays in the three cell types using this fusion protein to compare the resulting fluorescence data with the results of the semi-quantitative Western blots.

After detachment of cells from the cell culture dish using trypsin + EDTA, cells were transferred into a 2 ml reaction vessel and suspended in 1 ml of low bicarbonate saline (LBS) working solution (132.5 mmol/l NaCl, 4.8 mmol/l KCl, 1.2 mmol/l $MgCl_2 \cdot 6H_2O$, 1.2 mmol/l $KH_2PO_4$, 5.95 mmol/l HEPES (free acid), 9.05 mmol/l HEPES (sodium salt), 6 mmol/l D-glucose, 1,3 mmol/l $CaCl_2$). Cells were centrifuged at $600 \times g$ for 2 min at room temperature (RT). The supernatant was discarded and the cell pellet was resuspended in 1 ml 37˚C prewarmed LBS. A small part of the cell suspension was taken and stained with erythrosine B and cell count was determined using the LUNA™ Automated Cell Counter (Logos Biosystems, Villeneuve-d'Ascq, France). Cells in suspension were treated with 1815 ng/ml rHla-eGFP (corresponding to a concentration of 1,000 ng/ml of pure rHla) for 30 min on a shaker at 37˚C. Cells were again centrifuged at $600 \times g$ for 2 min at room RT, the supernatant was discarded and the cell pellet was resuspended in 1 ml LBS. The cell suspension (250 μl, each) was transferred into the wells of a black flat bottom chimney 96-well plate (CEK8.1, Roth Karlsruhe, Germany). The total fluorescence intensity of eGFP was determined using the plate reader Infinite® 200 PRO (Tecan, Männedorf, Switzerland) and the fluorescence intensity per cell was calculated. LBS was used as a blank.

Suspensions of cells treated as described above were also analysed for their GFP-fluorescence using a FACSAriaIIIu high-speed cell sorter (Becton Dickinson Biosciences, San Jose, CA, USA) with 488 nm excitation from a blue Coherent Sapphire solid state laser (18 mW). Optical filters were set up to detect the emitted GFP fluorescence signal at 530/30 nm (FITC channel). All fluorescence data were recorded at logarithmic scale with the FACSDiva8.02 software. Prior to measurement of experimental samples, the proper function of the instrument was checked using the cytometer setup and tracking software module (CS&T) together with CS&T beads (Becton Dickinson). First, in a SSC-area versus FSC-area dot plot the cell population was gated. The detection thresholds and photomultiplier (PMT) voltages were adjusted by using cells that had not been treated with rHla-GFP. The GFP signal from the scatter gate population was monitored in a GFP-area histogram. For each sample 10,000 events in the scatter gate were recorded.

### Determination of cell sizes

Cells were detached using trypsin + EDTA and suspended in 2 ml reaction vessels in LBS containing 0.01 mg/ml trypsin inhibitor. Volumes of 20 μl to 50 μl of these cell suspensions were

transferred into 10 ml of CASYton (OLS, Bremen, Germany) and the cell volumes of the individual cells were measured using the CASY Cell Counter + Analyzer (OLS, Bremen, Germany).

## Semi-quantitative FACS analysis

ADAM10 / α5β1-Integrin: This protocol was optimized for the quantification in intact cells of the number of relevant transmembrane proteins containing extracellular domains. Cells were incubated with accutase + EDTA at 37˚C until cells detached from the cell culture dish. Detached cells were transferred into a 2 ml reaction vessel and remaining cells were rinsed off the dish with 1 ml PBS containing 0.05% (w/v) sodium azide and transferred into the reaction vessel. Sodium azide inhibits the endocytic uptake of antibodies in the subsequent experimental steps [38]. Samples were centrifuged at $600 \times g$ for 5 min at RT, the supernatant was discarded and cells were resuspended in 1 ml PBS containing sodium azide and counted as described above. One million cells per sample were evenly distributed to two 1.5 ml reaction vessels. These tubes were centrifuged at $600 \times g$ for 5 min. To block non-specific binding sites of antibodies, cell pellets were resuspended in ice-cold blocking solution (3% bovine serum albumin (BSA) [w/v] in PBS with 0.05% [w/v] sodium azide) and incubated for 1 h at 4 C˚ on a shaker platform at low rotation speed. Cells were centrifuged at $600 \times g$ for 5 min, and cell pellets were resuspended in 100 μl antibody solution (1% BSA [w/v]; 0.1% saponin [w/v] in PBS containing 0.05% sodium azide [w/v]). Primary antibodies (2 μg anti-ADAM10 or anti-α5β1-integrin antibodies) were added and the tubes were incubated on a shaker platform for 1 h at RT. Subsequently, cells were washed 3 times with ice-cold PBS + sodium azide and pelleted at $600 \times g$ for 5 min. Cell pellets were resuspended in 100 μl secondary antibody solution containing anti-mouse Alexa 594 antibody (for ADAM10 primary antibody) (1:200) or anti-rabbit Alexa 594 antibody (for α5β1-integrin primary antibody) (1:500) and incubated on a shaker platform for 1 h at RT. Cells were washed 3 times with ice-cold PBS + sodium azide and centrifuged at $600 \times g$ for 5 min. The supernatants were discarded and cells were resuspended and fixed in 100 μl 4% paraformaldehyde (PFA) solution in PBS and incubated for 10 min at RT on a shaker platform. Cells were washed 3 times with PBS + sodium azide and pelleted at $800 \times g$ for 5 min. Finally, cells were resuspended in 200 μl PBS + sodium azide and stored in the dark at 4˚C before being analysed by flow cytometry as described above, but with PE-Texas Red optical filters.

Caveolin-1: This protocol was optimized for the quantification of the number of relevant membrane associated proteins that do not contain extracellular domains. In these experiments, we followed the protocol described above for ADAM10 or α5β1 integrin, but the washing steps were performed with normal PBS. Supernatants were discarded and cell pellets were resuspended in 4% PFA and incubated for 10 min at RT on a shaker platform before cells were washed 3 times with ice-cold PBS and sedimentation at $800 \times g$ for 5 min. To permeabilize fixed cells, cells were incubated in permeabilizing solution (0.1% saponin [w/v] in PBS) for 20 min at RT on a shaker platform. Cells were washed again 3 times with ice-cold PBS and centrifuged at $800 \times g$ for 5 min. Blocking of non-specific antibody bindings sites was performed for 45 min at RT on a shaker platform using 3% BSA (w/v) in PBS. Cells were centrifugated at $800 \times g$ for 5 min, the supernatants were discarded and cell pellets were resuspended in 100 μl antibody solution (anti-Caveolin-1 antibody (1 μg), 1% BSA [w/v]; 0.1% saponin [w/v] in PBS). Upon incubation (1 h, RT, shaking platform), cells were washed 3 times with ice-cold PBS and pelleted at $800 \times g$ for 5 min. Cells were then incubated with 100 μl secondary antibody solution (anti-mouse Alexa 594 antibody, 1:200) on a shaker platform for 1 h at RT. To remove unbound antibodies, cells were washed 3 times with ice-cold PBS and pelleted at $800 \times g$ for 5 min and stored in the dark for a short time at 4˚C before being analysed by flow cytometry as described above.

Cells used as negative controls were treated either with a mouse $IgG_{2b}$ antibody or with the respective secondary antibody. The remaining steps were the same as described above.

## Fluorescence microscopy

ADAM10 / α5β1-Integrin: Cells were cultured on 18 mm round coverslips in 12 well-plates. On the day of the experiment cells on the coverslips were washed 3 times with ice-cold PBS with 0.05% sodium azide (w/v) and then incubated with blocking solution (3% BSA in PBS containing sodium azide) for 45 min on ice. Cells were incubated with ADAM10 or α5β1-integrin primary antibody (both 1:50) at RT for 2 h, washed 3 times with ice-cold PBS with sodium azide and then incubated with the secondary antibodies (1: 200, anti-mouse Alexa 594 antibody for samples with ADAM10 primary antibody or 1:500, anti-rabbit Alexa 594 antibody for samples with α5β1-integrin primary antibody) for 1.5 h at RT. All antibodies were dissolved in PBS containing 1% BSA. For fixation, 4% PFA solution was added and coverslips were incubated at RT for 15 min. After washing the adherent cells on the coverslips, DAPI (500 ng/ml) was added as a nuclear counterstain. After repeated washing with PBS and double-distilled water, the coverslips were mounted to microscopic slides using Mowiol 4–88 and 1,4-diazabicyclo[2.2.2]octane (DAPCO). Slides were dried and stored at 4˚C overnight.

Caveolin-1: In this case, cells grown on coverslips as described above were first fixed using 4% PFA and then permeabilized with 0.1% saponin (w/v) in PBS. A caveolin-1 antibody (1:50) and an anti-mouse Alexa 594 antibody (1:200) dissolved in PBS containing 1% BSA (w/v) and 0.1% saponin (w/v) were used as primary or secondary antibodies, respectively.

Immunofluorescence images were taken using a LEICA DMi8 microscope (Leica Mikrosysteme Vertrieb GmbH, Wetzlar, Germany) equipped with a 63 x oil immersion objective. For storage and image processing the software LAS X (Leica) was used.

## Semi-quantitative analysis of plasma membrane phospholipids by high-performance thin-layer chromatography (HPTLC)

16HBE14o-, S9 or A549 cells were cultured to more than 90% confluency on 10 cm cell culture dishes, briefly washed with PBS, detached from dishes using trypsin + EDTA and suspended in pre-warmed (37˚C) LBS. Cells were centrifuged at 600 × g for 2 min at RT, resuspended in 1 ml LBS and counted as described above.

Cells were treated with ~ 140 pmol/ml *Bacillus cereus* sphingomyelinase or PBS at 37˚C on a shaker platform for 2 h. Lipids were extracted from $6 \times 10^6$ cells as described [39] using chloroform/methanol (ratio 2:1). Lipid extracts (40 μl per 6 x $10^6$ cells) were applied to a high performance thin-layer chromatography (HPTLC) silica 60 gel plate (Merck, Darmstadt, Germany) and lipids were separated using chloroform/methanol/water (65/25/4, v/v/v) as the mobile phase. To stain the separated phospholipids on the plate, a reagent containing ammonium heptamolybdate and tin(II)chloride were sprayed onto the HPTLC plate staining the phospholipids light blue. For the identification of lipids commercial standards of sphingomyelin, phosphatidylcholine, phosphatidylserine, and phosphatidylethanolamine were used. Images of the HPTLC plate with the stained lipids were taken immediately after staining and three days later, because sphingomyelin staining developed slowly. Phoretix 1 D was used for the densitometric analysis of the lipid spots.

## Data presentation and statistics

Original data are deposited at BioStudies (EMBL-EBI) (https://www.ebi.ac.uk/biostudies/) under the accession number S-BSST397. Data are presented as means ± S.D. of n experiments on different cell preparations. Significant differences in the series of means were detected by

ANOVA. Means were tested for significant differences to the appropriate controls using Student's t-test, if variances were equal, or otherwise Welch's t-test was used. Significant differences of means were presented as: $p < 0.05$ (*), $p < 0.01$ (**) and $p < 0.001$ (***).

## Results

### Differences in *S. aureus* alpha-toxin sensitivities of airway model epithelial cell lines

Previous investigations using an rHla concentration of 2,000 ng/ml to treat airway epithelial model cells (16HBE14o-, S9, A549) have revealed that all these cell types showed alterations in cell shape and the dissolution of cell-cell and focal contacts. However, contrary to 16HBE14o- or A549 cells which were damaged irreversible, the S9 cells were able to quickly recover from the effects of the toxin even in the continuous presence of the toxin in the cell culture medium [31, 34]. To get more detailed data on the relative alpha-toxin sensitivities of the model cells, we exposed the cells to 1,000 ng/ml rHla (30 nmol/l). This concentration results in specific binding of monomers to plasma membrane receptors and avoids non-specific binding at plasma membrane lipids [9]. Time lapse movies (one image every 3 min for 24 h) were generated and three time points (0, 3 and 12 h) were selected for detailed documentation of developing or closing gaps in the cell layers or for cells leaving the image plane indicating loss of cell matrix contact (Fig 1, upper panels).

All three cell lines reacted to the rHla concentration of 1,000 ng/ml (Fig 1). At 3 h of rHla-exposure, S9 cells exhibited the largest gaps in the cell layer, but recovered, so that the cell layer was completely closed again after 12 h. In addition, there were only very small numbers of cells detached from the culture plate at this time. The 16HBE14o- cells showed only little changes (low degree of gap formation in the cell layer) at 3 h exposure compared with the controls. However, at 12 h exposure, gap formation was clearly visible and many cells had lost contact with the culture plate. A549 cells seemed to react most sensitively to the toxin. After 3 h, gap formation was clearly visible and became most pronounced at 12 h of rHla exposure. In addition, many cells were detached from the culture plate at this time. The A549 and 16HBE14o- cells did not recover with respect to closing the cell layer.

### Differences in the binding capacity of Hla in airway epithelial model cells

A potential explanation for the differences in sensitivities against rHla of the airway epithelial model cells under investigation may be that the cells contain different amounts of bound mono- or heptamers at a given time of rHla exposure. To determine the amounts of bound rHla in all three cell lines (16HBE14o-, S9, A549) cells treated for 1 h with 1,000 ng/ml rHla, and whole cell protein extracts were analyzed by semi-quantitative Western blotting. In an alternative approach, cells were treated for 30 min with 28.9 pmol/ml of an rHla-eGFP fusion protein (corresponding to a concentration of 1,000 ng/ml rHla). Average fluorescence of bound toxin was determined with a microplate reader. Single cell fluorescence was measured using flow cytometry.

Both rHla-monomers and rHla-heptamers could be detected by Western blotting in all three cell types due to the SDS resistance of heptameric alpha-toxin (example blots in Fig 2A). While the variation of representative samples of the same cell line was quite low, the variation of samples of different cell lines was very high. Most rHla-heptamers were formed in the 16HBE14o- cells, less in the A549 cells and least in the S9 cells (Fig 2A). Densitometric analyses of Western blot images revealed that 16HBE14o- cells had incorporated approximately 10 times the number of rHla heptamers that were present in S9 cells and twice as much as in

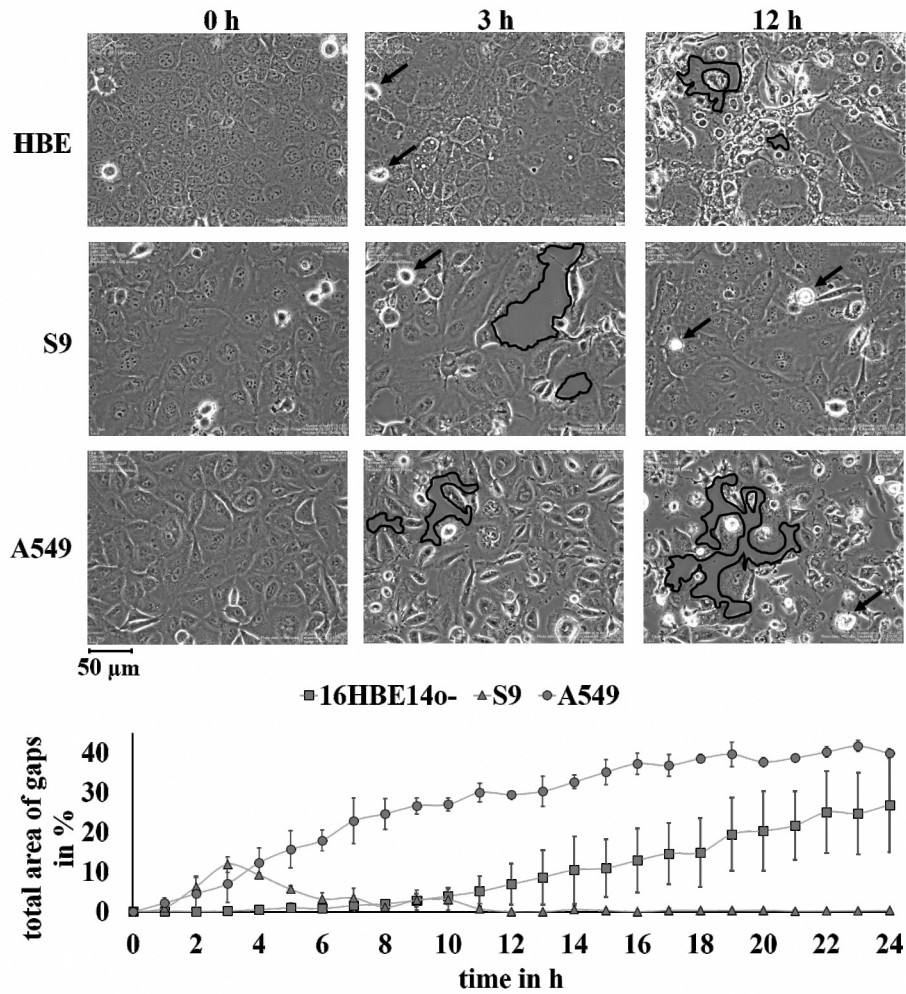

**Fig 1. Gap formation in cell layers and detachment of cells in confluent cultures of 16HBE14o-, S9 and A549 cells upon rHla exposure.** The upper panels show representative images from time lapse movies at 0, 3 or 12 h of rHla exposure. Black bordered areas indicate representative locations of paracellular gap formation in the cell layers. Black arrows point to examples of detached cells that have moved out of the focus plane. The lower panel shows time curves of gap formation in the cell layers derived from time lapse images obtained over 24 h of rHla exposure of cells at every full hour (means ± S.D.; n = 3).

A549 cells (Fig 2B). The amounts of cell-associated rHla monomers were somewhat variable (Fig 2C), but their pattern did not match the different levels of sensitivities in the three cell lines.

Analyses of total binding of rHla-eGFP mono- and heptamers to cells of the three lines confirmed the findings of the Western blots. Measurements of the total fluorescence intensities of bound rHla-eGFP using the plate reader (Fig 2D) revealed that 16HBE14o- cells bound approximately 10 times more fusion protein than S9 cells. A549 cells displayed approximately twice as much fluorescence as the S9 cells. Fluorescence data measured in individual cells using FACS analysis showed a similar pattern (Fig 2E). 16HBE14o- cells had the highest fluorescence levels followed by A549 cells (approx. 50% of that in 16HBE14o- cells) and S9 cells (less than 30% of that in 16HBE14o- cells).

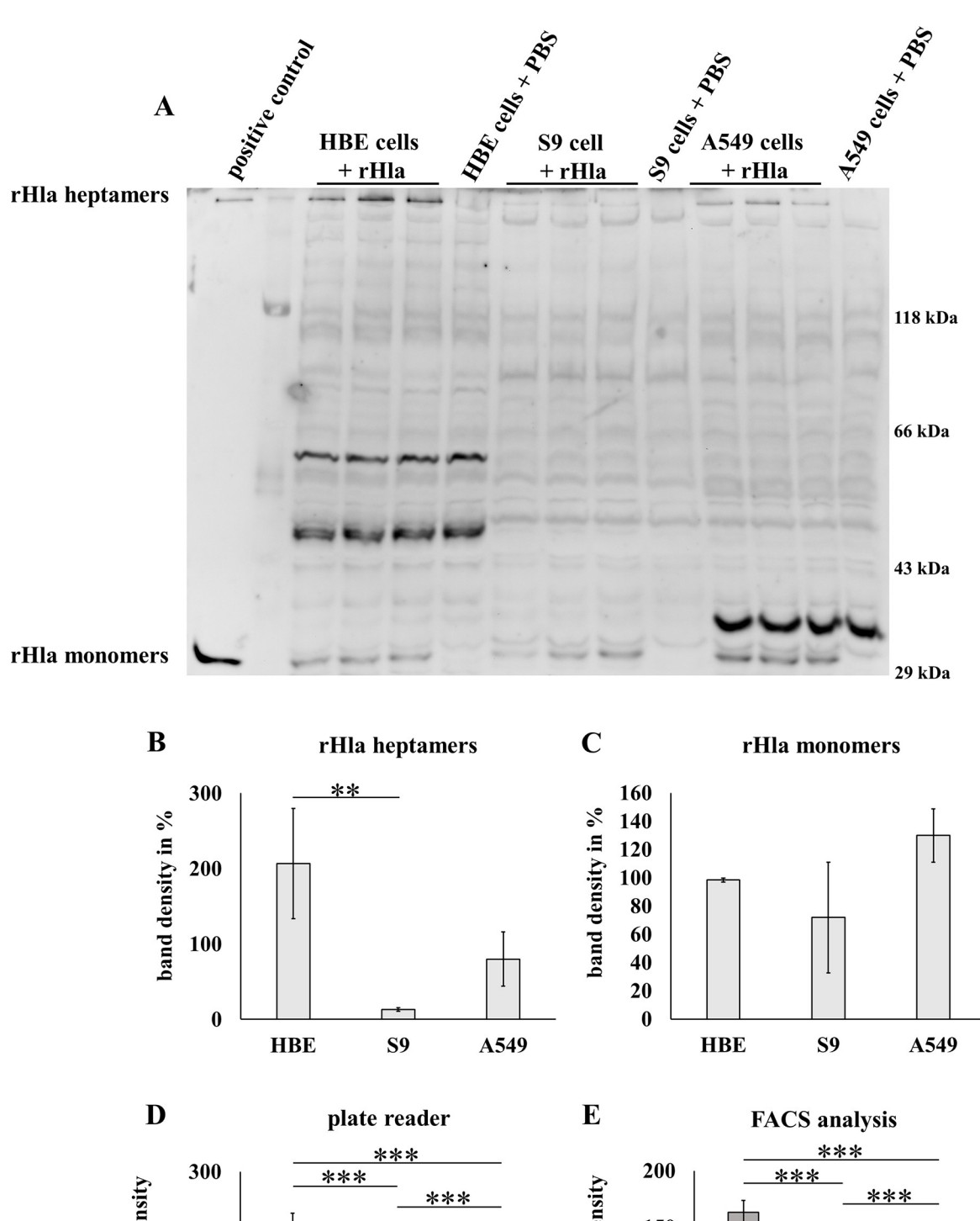

**Fig 2. Hla binding in three human airway epithelial model cell lines (16HBE14o-, S9 and A549) using Western blotting and fluorescence-based binding assays.** A-C: Confluent cell layers were treated with 1,000 ng/ml rHla or PBS as negative control for 1 h and whole cell lysates were used for Western blotting and examined for rHla mono- and heptamer bands. Recombinant rHla (20 ng per lane) was used as a positive control and showed a dense monomer band but also a faint heptamer band due to spontaneous multimerization of rHla monomers during storage (A, left lane). For each cell line, three preparations of cell extracts with equal total protein content were loaded in individual lanes of a single gel to compare signal intensities within and between cell lines (A). The Western blot image represents the results of 3 biological replicates. Densitometric analyses of SDS-resistant heptamer bands (B) or monomer bands (C) were done on 3 biological replicates (means ± S.D.). D-E: Cells in suspension were treated for 30 min with 1,815 ng/ml of eGFP-coupled rHla (corresponding to a concentration of 1,000 ng/ml of pure rHla). Total fluorescence intensity was determined using a plate reader (D, n = 4). In addition, the fluorescence intensity per cell was determined by flow cytometry, which is represented as the fluorescence intensity of the median of the measured cells (E, n = 3). Individual means were normalized to the total density of all Hla-bands in the respective membrane (B, C) or fluorescence intensity of all samples (D, E), presented in % and tested for significant differences using Student's t-test or Welch's t-test, respectively: * = $p \leq 0.05$, ** = $p \leq 0.01$ or *** = $p \leq 0.001$.

## Cell size

Cell size could be a decisive factor in determining the amount of Hla that bind to a given cell. The larger the cell surface, the more surface area exists for binding molecules and attachment sites of the toxin as well as for the formation of transmembrane pores. Thus, we measured the volumes of suspended cells of all three cell types using a CASY Cell Counter + Analyzer as these cells are rounded which gives a much better measure for cell size than the irregular shapes of attached cells within the cell layers. As shown in Fig 3, the cell volumes of suspended 16HBE14o- cells, S9 cells or A549 were between 3,500 and 4,200 fl. However, means of cell volumes did not significantly differ (means ± S.D., n = 4, each). However, means of cell volumes were only different between 16HBE14o- cells and A549 cells, but not between the S9 cells and the other two cell lines.

## Abundance of two potential Hla receptors and of caveolin-1 as a potential pore-stabilizing factor

To investigate the potential roles of ADAM10 or α5β1 integrin as plasma membrane receptors for rHla in the three lines of airway epithelial cells as well as the potential role of caveolin-1 as a stabilizer of the rHla-pore, we determined expression of these proteins in the three cell lines. As a first step, we optimized binding of the antibodies to their respective antigens in the extracellular or intracellular spaces. Primary and secondary antibodies were applied to the cells grown on coverslips prior to fixation for the detection of the extracellular domains of ADAM10 or α5β1 integrin, respectively. Primary and secondary antibodies for the detection of caveolin-1 were applied only after fixation of the cells on coverslips to allow contact with the intracellular epitope of this protein.

Immunefluorescence images of cell layers in which the nuclei were counterstained using DAPI showed that ADAM10 is expressed in all three cell lines (Fig 4A). In all cases, ADAM10 was mainly localized at the plasma membrane. A similar observation was made for α5β1 integrin. Caveolin-1, however, while being detected in all cell types, showed a somewhat different subcellular distribution with one portion attached to the plasma membrane, another portion distributed in particles in the cytosol (Fig 4A).

Analyses of suspended cells treated with antibodies against ADAM10, α5β1 integrin or caveolin-1 using flow cytometry (Fig 4B) revealed that 16HBE14o- cells had the highest expression level of ADAM10 followed by A549 cells (approx. 50% of that of 16HBE14o- cells) and S9 cells (approx. 30% of that of 16HBE14o- cells). A different result was obtained for α5β1 integrin that was highly expressed in S9 cells. In 16HBE14o- or in A549 cells, this protein showed only low expression levels (approx. 30% of the S9 level, each). Expression of caveolin-1 was equal in all three cell types (Fig 4B).

## Comparison of the amounts of cellular phospholipids as potential interaction sites for rHla

For semi-quantitative determination of phosphatidylcholine and sphingomyelin contents in 16HBE14o-, S9 or A549 cells, we extracted total lipids from confluent cell layers. Lipids of equal numbers of cells were then separated on HPTLC plates. Four phospholipids (sphingo-myelin (SM), phosphatidylcholine (PC), phosphatidylserine (PS) and phosphatidylethanol-amine (PE)) which are the most abundant ones in plasma membranes of eukaryotic cells [40] were used as standards. Phosphatidylcholine could not be entirely separated from phosphatidylserine (Fig 5A, panel a) so that both lipids were jointly quantified. All three phospholipids were present in the extracts prepared from the three cell lines (Fig 5A). Pre-treatment of intact cells with sphingomyelinase of *Bacillus cereus* resulted in complete elimination of sphingomye-lin from the cells (Fig 5A, panel b) confirming the identity of the respective phospholipid spots on the HPTLC plates. Densitometric analyses of the phospholipid spots (Fig 5B, left panel) showed that the highest abundance of sphingomyelin was present in 16HBE14o- cells while the other two cell types (S9, A549) had only half as much. Only marginal differences occurred in the combined amounts of phosphatidylcholine and–serine in the lipid extracts (Fig 5B, right panel). Again, 16HBE14o- cells carry most of this lipid material, followed by S9 and A549 cells. However, A549 cells had at least 70% of combined amounts of phosphatidylcholine and–serine in the lipid extracts compared with 16HBE14o- cells indicating that the differences in abundance of these lipids are not as large as those observed for sphingomyelin.

## Discussion

Differences in the relative sensitivities of eukaryotic host cells against pore-forming bacterial toxins [9, 31, 34] may have various reasons. The initial interaction of the toxin with the host

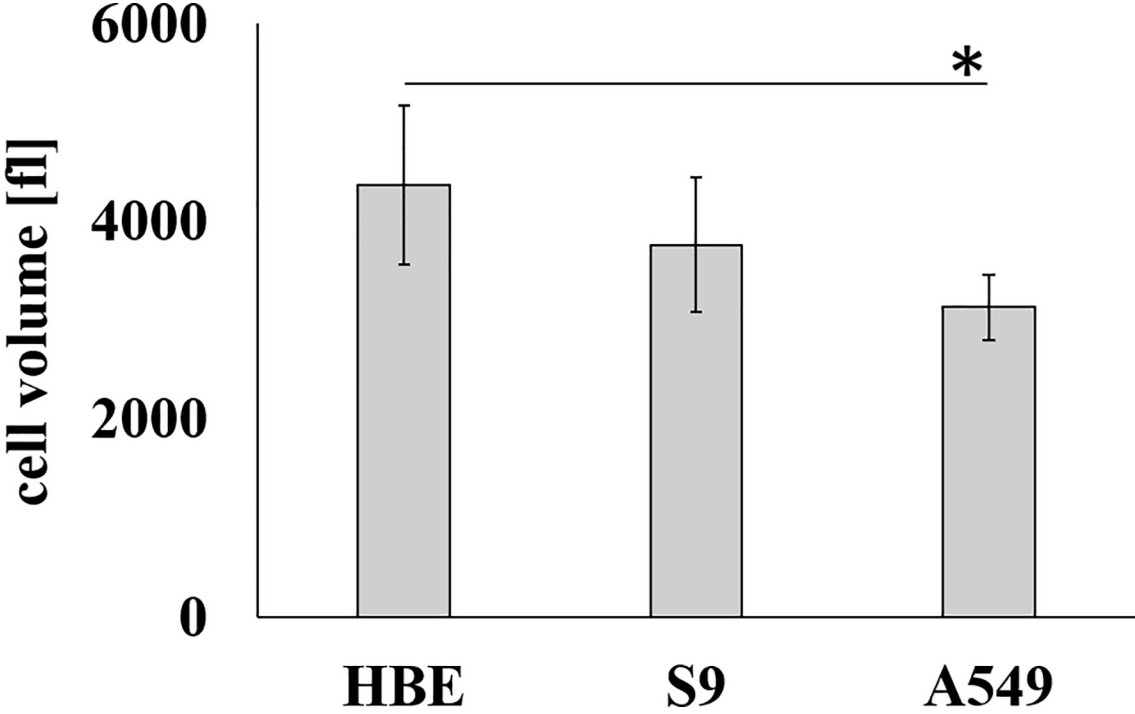

**Fig 3. Mean cell volumes of 16HBE14o-, S9 and A549 cells in suspension.** Individual means were tested for significant differences using Student's t-test: * = p ≤ 0.05 (means ± S.D.; n = 4).

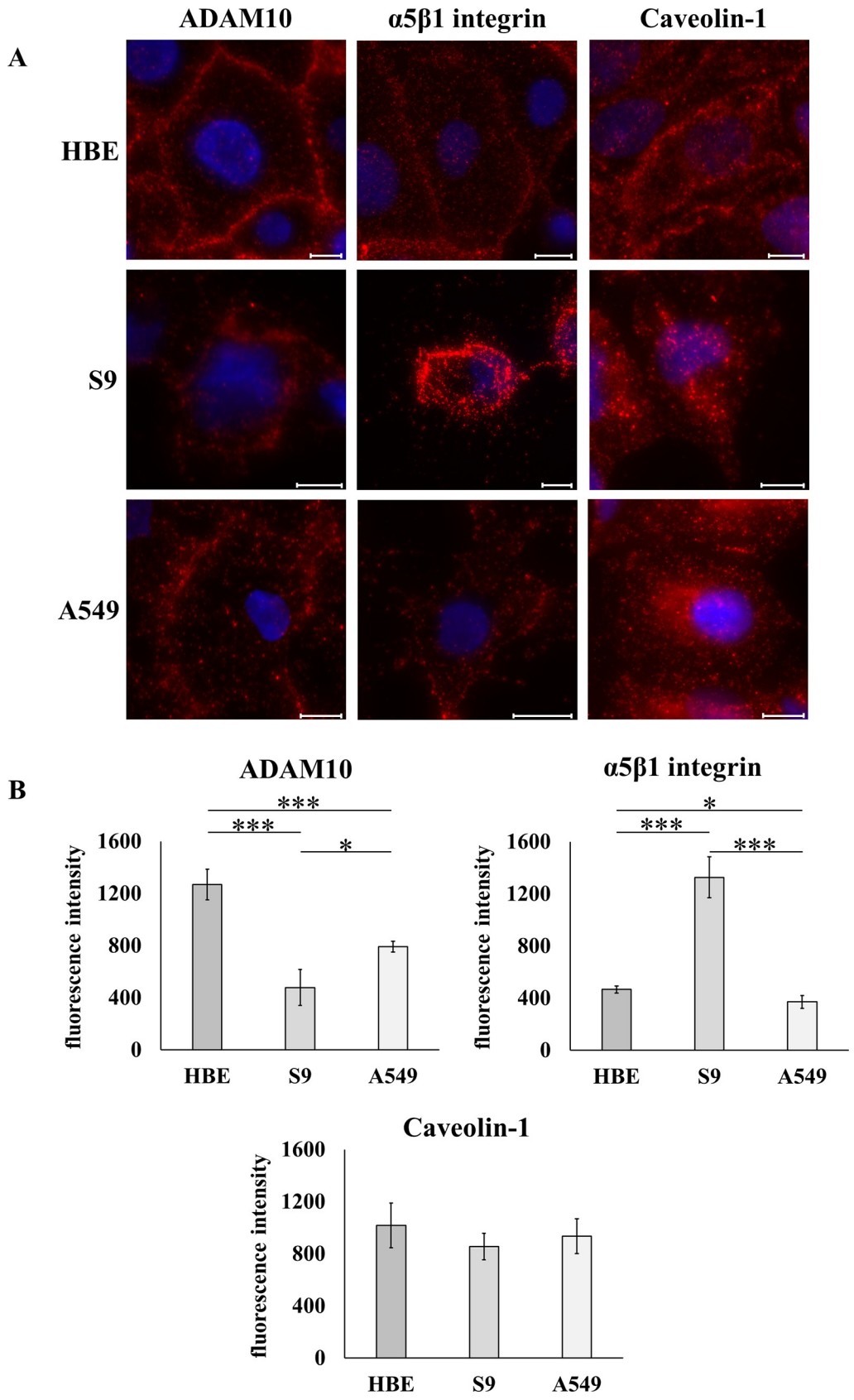

**Fig 4. Abundance of two potential Hla receptors and of caveolin-1 as a potential pore-stabilizing factor.** A: Representative examples of immune fluorescence assays using epifluorescence microscopy (nuclei counterstained using DAPI) that were performed on 16HBE14o-, S9 or A549 cells grown on coverslips using antibodies against ADAM10, the α5β1 integrin or against caveolin-1. Staining appearing in red represents specific labelling of the respective proteins. Scale bars: 10 μm. B: Semi-quantitative determination of primary and secondary antibody-mediated fluorescence in suspended individual cells by flow cytometry. During flow cytometry, the fluorescence of the antibody-tagged proteins per cell was measured and the respective medians of the detected peaks were used for calculating the means ± S.D. for the biological replicates (n = 4, each). Individual means were tested for significant differences using Student's t-test or Welch's t-test: * = p ≤ 0.05, ** = p ≤ 0.01 or *** = p ≤ 0.001.

cell seems to be facilitated by host cell receptor molecules that are utilized by the toxin to attach itself to the host cell surface. The monomeric alpha-toxin of *S. aureus* seems to bind with preference to ADAM10 (a disintegrin and metalloproteinase 10) associated with the outer leaflets of plasma membranes of many mammalian cells including human fibroblasts and epithelial cells [12, 13, 41, 42]. The amount of receptor molecules on a given cell may therefore determine the number of toxin molecules that simultaneously bind to the cell surface when the toxin concentration in the extracellular space is sufficiently high with respect to the affinity of the cell surface receptor. Cell size may be important as well as larger cells expressing receptor molecules at a certain density may bind a larger number of toxin molecules at a time compared with small cells. Furthermore, some researchers claim that the composition of the plasma membrane lipids may affect toxin binding [8, 10, 24]. Careful inspection of the literature, however, seems to indicate that the lipid composition of a given membrane is less involved in toxin binding but instead may enhance or attenuate the rate of multimerization of toxin monomers and the formation of the heptameric pre-pore. Especially lipids carrying a phosphocholine headgroup (sphingomyelin and phosphatidylcholine) as well as cholesterin have been implicated in favouring this process. Generally, pore formation is facilitated in membranes enriched in lipids that enhance membrane fluidity [43] but diminished in cells that are unable to synthetize sphingomyelin [42] or have been depleted of cholesterol or sphingomyelin [25, 44]. We reasoned that the amount of sphingomyelin that is present in the plasma membrane of a given cell type should substantially affect the rates of heptamerization and pore-formation and may also determine the sensitivity level of a cell toward alpha-toxin.

Stabilization of the pre-pore complex in the plasma membrane by accessory proteins may be another factor affecting cellular sensitivity to alpha-toxin. Caveolin-1 has been implicated in this context [18, 45].

The aim of this study was to analyze expression and abundances of potential plasma membrane receptors for alpha-toxin monomers (ADAM10, α5β1 integrin) as well as caveolin-1 as a potential pre-pore stabilizer in three different types of human airway epithelial cells (16HBE14o-, S9, A549). We compared these patterns with the patterns of sensitivity (measured as intensity of paracellular gap formation in rHla-treated cell layers) of these cell types toward alpha-toxin to find coincidences that pinpoint potentially relevant interaction mechanisms. We set out to apply the same approach to the composition of plasma membrane phospholipids in the three cell types.

We were able to confirm previous observations [31, 32, 34] that the effects of alpha-toxin on cell layers of S9 cells are less sustained than those on cell layers of 16HBE14o- and A549 cells (Fig 1). Gaps in the cell layers of S9 cells appeared between 2 and 6 h of toxin exposure and were completely closed at later times. Only a few S9 cells detached from the cell culture plate, but this occurred at the same rate in control cells not treated with rHla. However, gaps in the layers of 16HBE14o- cells appeared after 10 h of toxin exposure and cells were not able to reverse this effect. Layers of A549 cells behaved similarly, apart from the fact that gap formation started right after the addition of toxin. In both cases, a lot of cells detached from the

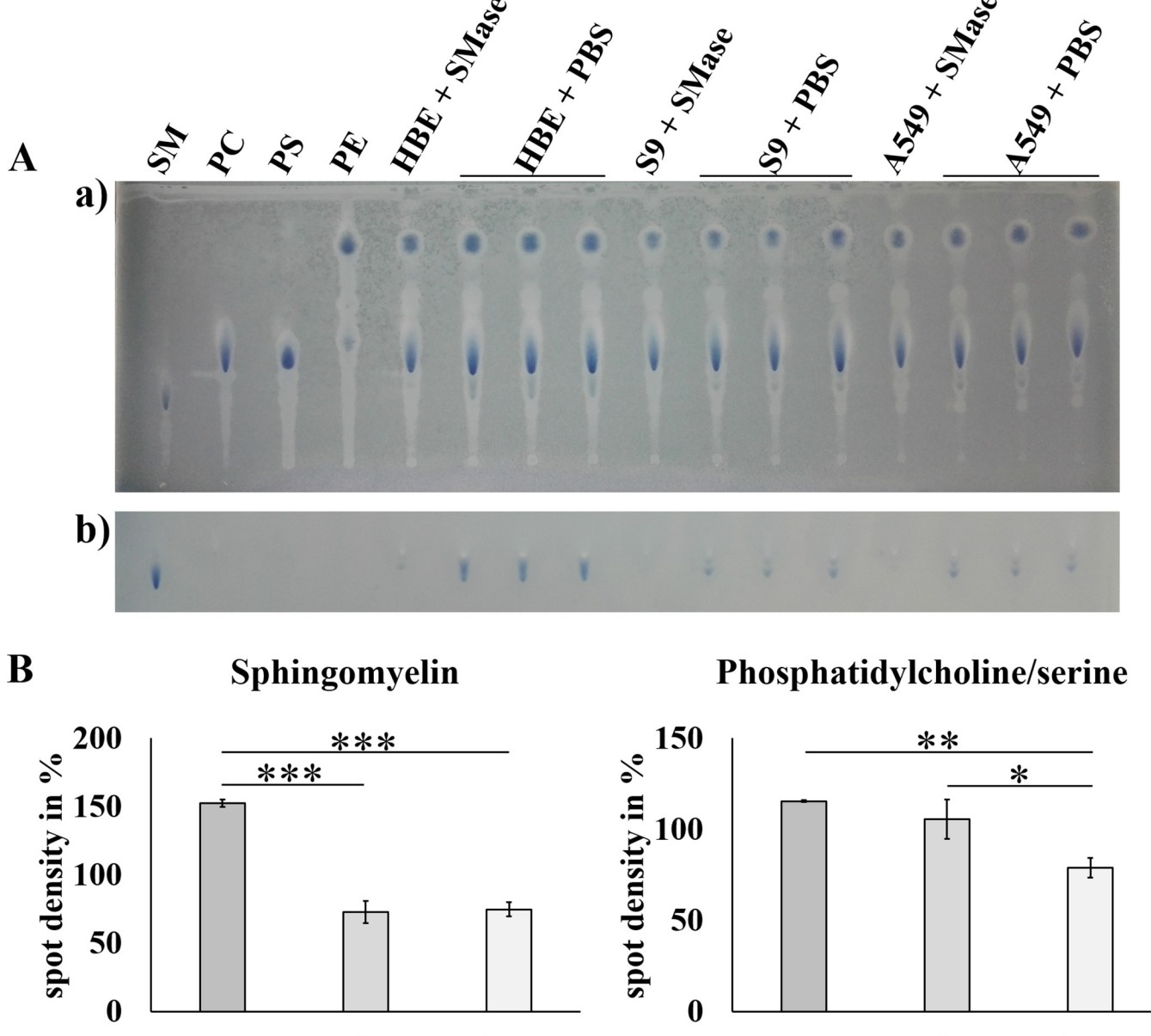

**Fig 5. Comparison of the amounts of cellular phospholipids as potential interaction sites for rHla.** Extracted lipids of the same number of cells were applied to the HPTLC plate, separated and stained with a reagent containing ammonium heptamolybdate. The plate was photographed directly after staining (A, panel a) and 3 days later (A, panel b), because only after 3 days the molybdenum blue develops full staining intensity with sphingomyelin. Commercial preparations of sphingomyelin (SM), phosphatidylcholine (PC), phosphatidylserine (PS) and phosphatidylethanolamine (PE) were used as controls. To confirm the identity of the sphingomyelin signal in the cell extracts, some cell preparations were pre-treated with a sphingomyelinase of *Bacillus cereus* (SMase). Densitometric analyses of the sphingomyelin spots (B, left panel) and the spot combinations of phosphatidylcholine and–serine (B, right panel) are reported as means ± S.D. (n = 3, each). Individual means were normalized to the total density of all spots of the respective phospholipid on the HPTLC plate and presented in % and tested for significant differences using Student's t-test or Welch's t-test: * = p ≤ 0.05, ** = p ≤ 0.01 or *** = p ≤ 0.001.

culture plate. This indicated that A549 cells are the most toxin-sensitive cells among those air-way epithelial cells tested in this study followed by 16HBE14o- cells. The least sensitive cells were the S9 cells (Fig 1).

This sensitivity sequence did not correlate at all with the binding of rHla monomers as detected by Western blotting (Fig 2C). The abundance of heptamers in rHla-treated cells, however, showed some similarities with the overall sensitivity pattern as S9 cells had the least number of heptamers upon incubation of cells with 1,000 ng/ml rHla for 1 h compared with 16HBE14o- and A549 cells. The latter showed higher band densities than S9 cells (Fig 2B) although the difference between S9 cells and A549 cells was statistically not significant. Measurements of total rHla-eGFP binding using a plate reader (Fig 2D) or of rHla-eGFP binding per cell using flow cytometry (Fig 2E) showed generally the same pattern as the Western blot assays. Binding was lowest in S9 cells, significantly higher in A549 cells and even more pronounced in 16HBE14o- cells. Cell volumes cannot account for these differences as cell volumes of the three cell types did not show a respective pattern (Fig 3). These results allow the conclusion that differences in monomer binding are probably not a major factor responsible for the observed differences in overall sensitivities of the cell types toward rHla. It seems likely that differences in the formation rate of heptamers from the pool of bound monomers or stabilization of heptamers in the plasma membranes may be more important.

Whether overall sensitivity of cells correlated with the abundance of endogenous, potentially rHla binding proteins was tested using a flow cytometric approach (Fig 4B) after checking for protein expression by immune fluorescence (Fig 4A). All three of the potential rHla binding proteins were shown to be expressed in 16HBE14o-, S9 as well as in A549 cells. Flow cytometric analyses of cell surface abundance of ADAM10 or of α5β1 integrin showed entirely different patterns. While the pattern of ADAM10 abundance correlated well with the total binding of toxin to the cells, the abundance of α5β1 integrin did not at all correlate with the results of the binding assays reported in Fig 2 or gap formation as a measure for cellular sensitivity against alpha-toxin (Fig 1). This indicates that ADAM10 may facilitate toxin monomer binding and heptamer formation in airway epithelial cells and confirms results presented previously [12, 46]. We could, however, not confirm previous results in other cell types that caveolin-1 plays a major role in pore-formation and overall sensitivity of cells to alpha-toxin [12, 18, 47]. As shown in Fig 4B, the abundance of caveolin-1 was equal in all three cell types. This does, however, not rule out a general function of caveolin-1 in stabilizing heptamers in these cells.

The combined spot volumes on HPTLC plates of phosphatidylcholine and phosphatidylserine in 16HBE14o-, S9 and A549 cells did not show any characteristic patterns that would have been compatible with the conclusion that these lipids may be instrumental in monomer binding or multimerization of alpha-toxin (Fig 5B, right panel). The differences in sphingomyelin abundances, however, show a pattern (Fig 5B, left panel) that may allow the conclusion that sphingomyelin is instrumental in heptamer formation [25]. It is definitely not consistent with the hypothesis, that sphingomyelin facilitates toxin monomer binding to airway epithelial cells.

Despite the high degree of monomer binding in A549 cells (Fig 2C), the abundance of heptamers in this cell type (Fig 2B) is rather low. In light of the above hypothesis, this may be explained by a relatively low rate of heptamer formation in the initial dynamic period of toxin exposure because the abundance of sphingomyelin, the lipid facilitating heptamerization, is low (Fig 5B, left panel). The same context may explain the very low abundance of heptamers in S9 cells (Fig 2B, left panel) although in this case, the somewhat lower level of monomer binding (Fig 2C) due to the low expression level of ADAM10 (Fig 4B, left panel) may contribute to this condition as well. 16HBE14o- cells have the highest expression level of ADAM10 of all three cell types. This may rapidly bring large numbers of alpha-toxin monomers in close vicinity to each other. If ADAM10 contributes to heptamerization as does sphingomyelin, which is highly abundant in these cells (Fig 5B, left panel), one would expect that toxin exposure results in

high levels of toxin heptamers in the cell membrane of these cells which is actually the case (Fig 2B). This may at least explain the much higher overall sensitivity of 16HBE14o- cells toward alpha-toxin in comparison with that of S9 cells. Why A549 cells seem to be the most sensitive of the cell lines tested in this study despite the fact that heptamer abundance is lower than that in 16HBE14o- cells remains unexplained. A549 cells may be more sensitive to the immediate cell biological consequences of pore-formation [3] so that already a small number of functional pores may have large impact on cellular fitness. Due to the relatively low sphingomyelin content (Fig 5B, left panel) more monomeric rHla may have remained at the plasma membrane of A549 cells and assembly of heptamers may have been delayed in comparison with 16HBE14o- cells which had less monomers and higher amounts of heptamers (Fig 2B and 2C). Furthermore, there may be an as yet unknown binding molecule in these cells that facilitates monomer binding and that the combination of low levels of sphingomyelin and phosphatidylcholine (Fig 5B) may somewhat attenuate the rate of heptamer formation and thus limit heptamer abundance (Fig 2B). Although the amounts of sphingomyelin were not significantly different in A549 and S9 cells (Fig 5B, left panel), A549 cells showed a significantly higher density of ADAM10 (Fig 4). The higher density of this putative Hla receptor in A549 cells may explain the fact that the amounts of bound rHla monomers were higher in A549 than in S9 cells (Fig 2C). This shows that, although heptamerization and pore formation may be more dependent on the abundance of sphingomyelin (c.f. Fig 2B), there may be also a certain contribution of receptor abundance on the rate of pore formation.

Another factor that could make S9 cells more resistant to alpha-toxin is the fact that these cells are able to increase their glycolytic activity under toxin exposure [34] in order to better compensate for the loss of ATP that diffuses through the pores into the extracellular medium [29]. 16HBE14o- cells, however, are not able to increase glycolytic activity during toxin exposure [34] which probably makes it more difficult for these cells to recover from the cellular effects of pore formation. That S9 cells quickly recover from the effects of toxin-mediated pore formation (Fig 1) may also depend on their ability to internalize functional pores via endocytosis and thus remove them from their plasma membranes. Such a defense mechanism has already been observed in other cell types [48, 49] and is currently under study in our model cells.

In summary, our data are consistent with the hypothesis that ADAM10 is the most important factor for facilitating alpha-toxin monomer binding to the surfaces of airway epithelial cells. ADAM10 may also play a permissive role in heptamer formation. The main factor accelerating heptamer formation out of the pool of plasma membrane bound monomers, however, seems to be sphingomyelin, potentially with smaller contributions of phosphatidylcholine. Nevertheless, there seem to be additional factors that affect the sensitivities of different airway epithelial cells to the toxin.

## Supporting information

**S1 File.**
(PDF)

**S1 Fig.**
(PDF)

**S2 Fig.**
(PDF)

**S3 Fig.**
(PDF)

**S4 Fig.**
(PDF)

## Acknowledgments

The authors thank Elvira Lutjanov and Dana Sponholz for excellent technical assistance. We are grateful to Yurong Ling for sharing the results of her experiments showing that rHla-eGFP is able to form functional pores in the plasma membranes of airway epithelial cells. We thank Prof. Dr. Ritter and Linus Gohlke from the Institute of Pharmacy, University of Greifswald for allowing us to use their fluorescence microscope and for sharing some reagents. None of the authors declares any conflict of interest. MSc Nils Möller is the recipient of a graduate student stipend (Landesgraduiertenstipendium) of the State of Mecklenburg-Vorpommern, Germany.

## Author Contributions

**Conceptualization:** Nils Möller, Sabine Ziesemer, Jan-Peter Hildebrandt.

**Data curation:** Nils Möller.

**Investigation:** Nils Möller, Sabine Ziesemer, Jan-Peter Hildebrandt.

**Methodology:** Sabine Ziesemer, Petra Hildebrandt, Nadine Assenheimer, Uwe Völker, Jan-Peter Hildebrandt.

**Project administration:** Jan-Peter Hildebrandt.

**Resources:** Jan-Peter Hildebrandt.

**Supervision:** Jan-Peter Hildebrandt.

**Validation:** Nils Möller, Sabine Ziesemer, Petra Hildebrandt, Nadine Assenheimer, Jan-Peter Hildebrandt.

**Writing – original draft:** Nils Möller.

**Writing – review & editing:** Sabine Ziesemer, Uwe Völker, Jan-Peter Hildebrandt.

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
