## [Decision Letter · Decision Letter 0]

7 Apr 2020

PONE-D-20-07341

S. aureus alpha-toxin monomer binding and heptamer formation in host cell membranes – Do they determine sensitivity of airway epithelial cells toward the toxin?

PLOS ONE

Dear Dr. Hildebrandt,

Thank you for submitting your manuscript to PLOS ONE. After careful consideration, we feel that it has merit but does not fully meet PLOS ONE’s publication criteria as it currently stands. Therefore, we invite you to submit a revised version of the manuscript that addresses the points raised during the review process.  Can the authors present a cell death analysis in which they establish an LC50 value based on a dose response curve showing percent cell mortality with varying concentrations of alpha-toxin?  Responding to the reviewer's comments in a direct manner would enhance the quality of the presentation.

We would appreciate receiving your revised manuscript at your earliest convenience.  To enhance the reproducibility of your results, we recommend that if applicable you deposit your laboratory protocols in protocols.io, where a protocol can be assigned its own identifier (DOI) such that it can be cited independently in the future. For instructions see: http://journals.plos.org/plosone/s/submission-guidelines#loc-laboratory-protocols

We look forward to receiving your revised manuscript.

Kind regards,

Lee Bulla, Jr.

Academic Editor

PLOS ONE

Additional Editor Comments (if provided):

The manuscript reports good science and, with some additional information, would be worthy of further consideration. Can the authors present a cell death analysis in which they establish an LC50 value based on a dose response curve showing percent cell mortality with varying concentrations of alpha-toxin? Responding to the reviewer's comments in a direct manner would enhance the quality of the presentation.

2. At this time, we ask that you please provide scale bars on the microscopy images presented in FIGURES and refer to the scale bar in the corresponding Figure legend.

3. Please provide additional information about each of the cell lines used in this work, including quality control testing procedures (authentication and characterisation). For more information, please see http://journals.plos.org/plosone/s/submission-guidelines#loc-cell-lines.

Reviewers' comments:

Reviewer's Responses to Questions

**Comments to the Author**

1. Is the manuscript technically sound, and do the data support the conclusions?

Reviewer #1: No

2. Has the statistical analysis been performed appropriately and rigorously? 

Reviewer #1: Yes

3. Have the authors made all data underlying the findings in their manuscript fully available?

Reviewer #1: Yes

4. Is the manuscript presented in an intelligible fashion and written in standard English?

Reviewer #1: Yes

5. Review Comments to the Author

Reviewer #1: Alpha-toxin is an important virulence factor of Staphylococcus aureus. Although Its receptor was identified as ADAM10, other proteins or membrane lipids were claimed to bind to alpha-toxin or stabilizes the toxin complex. In their previous study, the authors showed that the human respiratory epithelial cell lines, 16HBE14o- (HBE) and S9, and the lung cancer line A549 show different sensitivity to alpha-toxin. In this study, with those cell lines and using paracellular gap formation as the toxicity indicator, the authors sought to identify the factors determining the sensitivity to alpha-toxin. In the paracellular gap formation assay, as reported previously, A549 showed the highest sensitivity. HBE also showed significant gap formation. On the other hand, S9 showed gap formation at 3 h, but later the gap was completely repaired. In the alpha-toxin binding assay, unlike the gap formation assay, HBE showed the highest binding, followed by A549. S9 showed the lowest binding. Cell volume was not significantly different among the cell lines. When the abundance of the receptor proteins was measured, ADAM10 showed a good correlation with alpha-toxin binding: HBE showed the highest amount of ADAM10, followed by A549 and S9. Other proteins, such as alpha5beta1-integrin and caveolin-1, did not show any correlation with alpha-toxin binding. When membrane lipid composition was analyzed for the cell lines, the sphingomyelin level was highest in HBE, whereas it was not significantly different between A549 and S9. The level of phosphatidylcholine/serine mix showed no correlation with either paracellular gap formation or alpha-toxin binding. Base on these results, the authors concluded that the abundance of ADAM10 correlated best with gap formation or cell sensitivities, and the relative abundance of sphingomyelin in the plasma membrane might be used as an indicator for the sensitivity to alpha-toxin.

Considering the fact that ADAM10 is the receptor for alpha-toxin, it is not surprising to see the correlation between the abundance of ADAM10 and alpha-toxin binding. However, it is a significant contribution of the study to show that other proteins (i.e., alpha5beta1-integrin and caveolin-1), which were claimed to be a receptor for alpha-toxin, did not show any correlation with either alpha-toxin binding or gap formation. Unfortunately, I do not think the authors’ conclusion is supported by the results presented. For example, although HBE binds to alpha-toxin at least twice more than A549 does (Fig. 2), it showed much a lower paracellular gap formation than A549 (Fig. 1). The role of sphingomyelin in the sensitivity to alpha-toxin is not clear either. Fig. 5 shows that A549 and S9 contain similar levels of sphingomyelin; however, they show drastically different gap formation (Fig. 1).

Major comments:

1. Is it possible that the paracellular gap formation is not a good indicator for alpha-toxin sensitivity? I wonder if cell-death can be a better indicator for alpha-toxin sensitivity. The authors can treat the cell suspensions with various concentrations of alpha-toxin and measure the death of the cells by the Live/Death assay.

2. In Fig.4, Western blotting of the cell lysate can give more quantitative measurement for the receptor abundance.

Minor comments:

Line 40: proxi -> proxy

Line 390: cells treated -> cells were treated

Figure 5: The letters are shown in low quality.

6. PLOS authors have the option to publish the peer review history of their article (what does this mean?). If published, this will include your full peer review and any attached files.

Reviewer #1: No

---

## [Author Response · Author response to Decision Letter 0]

7 May 2020

PONE-D-20-07341, Revision

Responses to the Editor’s and Reviewer’s Comments and Changes made to the Manuscript

Editor: Please ensure that your manuscript meets PLOS ONE's style requirements, including those for file naming. The PLOS ONE style templates can be found at http://www.plosone.org/attachments/PLOSOne_formatting_sample_main_body.pdf and http://www.plosone.org/attachments/PLOSOne_formatting_sample_title_authors_affiliations.pdf

Authors: Unfortunately, these URLs do not work, but we have tried to match the file name requirements of the journal anyway.

Editor: At this time, we ask that you please provide scale bars on the microscopy images presented in FIGURES and refer to the scale bar in the corresponding Figure legend.

Authors: This applies to the images shown in Fig. 4. We have included the scale bars in every single image and referred to these bars in the legend to Fig. 4.

Editor: Please provide additional information about each of the cell lines used in this work, including quality control testing procedures (authentication and characterisation). For more information, please see http://journals.plos.org/plosone/s/submission-guidelines#loc-cell-lines.

Authors: The cell lines used in this study are well established standard cell lines of human airway epithelial cells that have been used in many previous studies by other authors and by us. Their characteristics have been described in many publications which have been cited in the Supplemental Information 1 to this manuscript.

Editor: PLOS ONE now requires that authors provide the original uncropped and unadjusted images underlying all blot or gel results reported in a submission’s figures or Supporting Information files …

Authors: All original image files as well as the entire set of experimental data relevant to this publication have been provided as Supplementary Information 2. Statistics are reported in Supplemental Information 3.

Reviewer: … Unfortunately, I do not think the authors’ conclusion is supported by the results presented. For example, although HBE binds to alpha-toxin at least twice more than A549 does (Fig. 2), it showed much a lower paracellular gap formation than A549 (Fig. 1). The role of sphingomyelin in the sensitivity to alpha-toxin is not clear either. Fig. 5 shows that A549 and S9 contain similar levels of sphingomyelin; however, they show drastically different gap formation (Fig. 1).

Authors: We do not claim that the densities of potential Hla receptors in the three cell types are the only determinants of their Hla sensitivity. Likewise, we do not say that the abundance of sphingomyelin is the only factor affecting sensitivity of cells toward Hla. Our point is that higher densities of potential receptors may result in more effective binding of Hla to the cell surface while formation of Hla heptamers and functional pores are facilitated by higher abundances of sphingomyelin. Combinations of these factors with the (so far not investigated) abilities of the cells to deal with the cell physiological effects or to remove functional pores from the cell surface may actually explain the different sensitivities in the airway epithelial cells types.

We have added three sentences to the discussion to explain these considerations more thoroughly.

Reviewer: Is it possible that the paracellular gap formation is not a good indicator for alpha-toxin sensitivity? I wonder if cell-death can be a better indicator for alpha-toxin sensitivity. The authors can treat the cell suspensions with various concentrations of alpha-toxin and measure the death of the cells by the Live/Death assay.

Authors: The aim of this experiment was not to test the lethality of the toxin, but to compare the cell physiological effects in the three cell lines to sub-lytic concentrations of Hla and to test whether the cells are able to recover from these effects. Thus, gap formation in the cell layers, which is a hallmark of all adverse effects of Hla on epithelial cells as we have shown in several previous studies, appears to be a much better measure for cell sensitivities of different cell types toward Hla compared with survival assays. It could be shown that A549 and S9 cells reacted to toxin exposure within a short time, but only S9 cells were able to recover and form confluent cell layers again. 16HBE14o- cells, however, showed gap formation only after longer exposure times, but were unable to recover. A simple live/dead-assay would not be able to pick up these subtle differences in cell layer performance under Hla, neither the time component nor the recovery component.

Moreover, significant effects in live/dead-assays can only be achieved when cells are treated with higher concentrations of Hla than those that have been used in this study. As shown previously, even the Hla concentration of 2,000 ng/ml was sub-lytic and did not significantly increase the number of dead cells (e.g. Ziesemer et al. 2017, DOI: 10.1165/rcmb.2016-0207OC). In the experiments of this study, however, we used the lowest possible concentration of Hla that would still trigger cell physiological effects in these cells and avoid any non-specific binding of the toxin to the plasma membrane in the binding assays. For this purpose, pre-tests were performed and based on these data we selected the optimal concentration of 1000 ng/ml rHla. 

Reviewer: In Fig.4, Western blotting of the cell lysate can give more quantitative measurement for the receptor abundance.

Authors: It is correct that Western blots allow quantitative measurements of receptor density. However, in these experiments the selection of the appropriate control protein for normalization is critical. Based on our previous proteome studies we know that the protein composition of these three cell lines differs substantially (Palma Medina et al. 2019 Mol Cell Prot 18, 892-908; Surmann et al. 2015 J Prot 128, 203-217, and unpublished data). For example, the amounts of the housekeeping protein β-actin differ in the three cell lines. Thus, normalization of the receptor protein bands to β-actin or other potential housekeeping proteins did not seem suitable. The FACS method, however, overcomes this problem due to the fact that the total receptor number per cell is determined and can be compared between the three cell types.

---

## [Editor Report · Decision Letter 1]

14 May 2020

S. aureus alpha-toxin monomer binding and heptamer formation in host cell membranes – Do they determine sensitivity of airway epithelial cells toward the toxin?

PONE-D-20-07341R1

Dear Dr. Hildebrandt,

We are pleased to inform you that your manuscript has been judged scientifically suitable for publication and will be formally accepted for publication once it complies with all outstanding technical requirements.

With kind regards,

Lee Bulla, Jr.

Academic Editor

PLOS ONE
---

## [Editor Report · Acceptance letter]

20 May 2020

PONE-D-20-07341R1 

*S. aureus* alpha-toxin monomer binding and heptamer formation in host cell membranes – Do they determine sensitivity of airway epithelial cells toward the toxin? 

Dear Dr. Hildebrandt:

I am pleased to inform you that your manuscript has been deemed suitable for publication in PLOS ONE. Congratulations! Your manuscript is now with our production department. 

With kind regards,

on behalf of

Dr. Lee Bulla, Jr. 

Academic Editor

PLOS ONE